

# Applying pySTEPS optical flow algorithms to improve convection nowcasting over the Maritime Continent

Joseph Smith[1], Cathryn Birch[1], John Marsham[1], Simon Peatman[1], Massimo Bollasina[2], George Pankiewicz[3]

[1]School of Earth and Environment, University of Leeds, Leeds, LS2 9JT
[2]School of GeoSciences, University of Edinburgh, Edinburgh, EH8 9YL
[3]UK Met Office, Exeter, EX1 3PB

*Correspondence to:* Joseph Smith (eejasm@leeds.ac.uk)

**Abstract.** The Maritime Continent (MC) regularly experiences powerful convective storms that produce intense rainfall, flooding and landslides, which numerical weather prediction models struggle to forecast. Nowcasting uses observations to make more accurate predictions of convective activity over short timescales (~0-6 hours). Optical flow algorithms are effective nowcasting methods as they are able to accurately track clouds across observed image series and predict forward trajectories. Optical flow is generally applied to weather radar observations, however, the radar coverage network over the MC is not complete and the signal cannot penetrate the high mountainous regions. In this research, we apply optical flow algorithms from the pySTEPS nowcasting library to satellite imagery to generate both deterministic and probabilistic nowcasts over the MC. The deterministic algorithm shows skill up to 4 hours on spatial scales of 10 km and coarser, and outperforms a persistence forecast for all lead times. Lowest skill is observed over the mountainous regions during the early afternoon and highest skill is seen during the night over the sea. A key feature of the probabilistic algorithm is its attempt to reduce uncertainty in the lifetime of small scale convection. Composite analysis of 3-hour lead time nowcasts, initialised in the morning and afternoon, show it produces reliable ensembles but with an under-dispersive distribution, and produced area under the curve scores (i.e. ratio of hit rate to false alarm rate across all probability thresholds) of 0.80 and 0.71 over the sea and land, respectively. When directly comparing the two approaches, the probabilistic nowcast shows greater skill at ≤ 60 km spatial scales, whereas the deterministic nowcast shows greater skill at larger spatial scales ~200 km. Overall, the results show promise for the use of pySTEPS and satellite retrievals as an operational nowcasting tool over the MC.

Keywords: Nowcasting, convection, pySTEPS

## 1 Introduction

The Early Warnings For All Initiative was launched by the United Nations in November 2022 and calls for the whole world to be covered by early warning systems by the end of 2027 ("World Meteorological Organization," 2023). It focuses on poorer countries in Asia, Africa, South and Central America and the Pacific, and motivates the development of early warning weather systems for these regions.

The Maritime Continent (MC) is a region of Southeast Asia that includes the countries of Indonesia, Malaysia, Philippines, Papua New Guinea, Brunei and East Timor. It is a complex mix of land and ocean with major islands such as Sumatra, Java, Borneo and New Guinea making it the largest archipelago on Earth (Figure 1a). It is also one of the wettest places on Earth with its complex topography and location across the equator making it a hotspot for extreme weather. The region often experiences natural disasters such as flooding and landslides that have disastrous effects on already very poor areas. Easterly trade winds blow warm water across the Pacific into the MC creating a 'warm pool' around the region (Dayem et al., 2007), which, when combined with its proximity to the equator and the inter-tropical convergence zone, provides favourable conditions for deep convection. The large amounts of latent heat released from this convection means that the region is often referred to as the 'boiler box' of the tropics as it plays a crucial role in contributing to the global atmospheric circulations (the Hadley and Walker Cells), in turn affecting both local and global weather systems (Ramage, 1968).





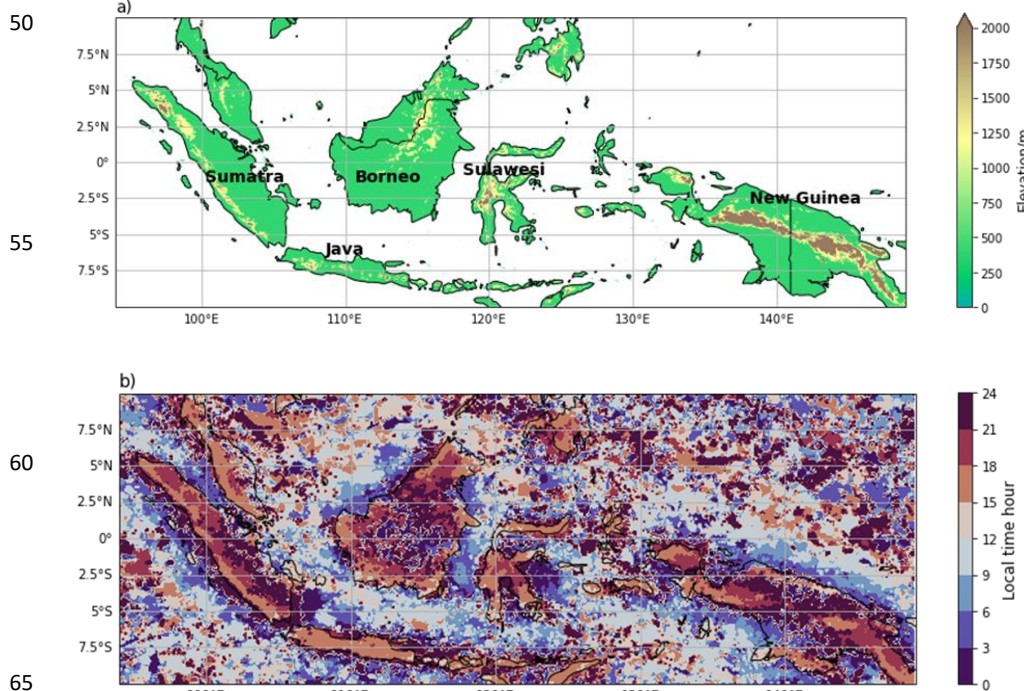

**Figure 1.** Maps of the selected islands within the MC showing a) the orography and b) the diurnal cycle of peak rainfall (interpolated to local solar time) using the Global Precipitation Measurement dataset (Hou et al., 2014) from December, January and February 2001 – 2020.

The MC's strong diurnal cycle (shown in Figure 1b by the spatial variation in timings of peak rainfall) is one of its dominant drivers of convective activity (Yamanaka, 2016). It has typical characteristics of most diurnal cycles across the tropics (Yang and Slingo, 2001), starting with peak solar insolation around midday. This starts to form a land-sea temperature contrast due to the lower heat capacity of the land. A sea breeze then develops blowing on land, often triggering convection which builds into the late afternoon and evening. Many islands in the MC also contain mountains close to the coast with altitudes of over 2000 m (e.g. Sumatra). Orographic lifting driven by the mountains can further enhance the convection (Mori et al., 2004). Into the late evening and overnight this convection propagates offshore until the early morning the following day, leaving clear skies over land in the morning for strong solar insolation to restart the process.

Numerical weather prediction (NWP) models struggle to represent the moist convection that dominates weather in the MC, with coarser resolution models often initiating convection too early in the day (Porson et al., 2019) or under-estimating the amount of rainfall (Qian, 2008). Ferrett et al., (2021) shows that the skill of ensemble forecasts from a higher-resolution, convective-scale configuration of the Met Office Unified Model, with 4.5 km horizontal grid spacing over Indonesia, only start to show skill (during the first day after initialisation) when coarsened up to spatial scales of ~150 km. Mesoscale convective systems are defined as having spatial scales of at least ~100 km and so convective-scale models cannot be relied upon to skilfully resolve impactful storms over the MC.

Nowcasting is the process of obtaining current observations of the atmosphere and using them to generate rapid, short-term (typically ~0-6 hours ahead) predictions of the future atmospheric state (Roberts et al., 2022). It requires real time observations (e.g. from weather radar) as an input and the application of predictive techniques



to forward propagate these observations. Unlike NWP models, nowcasting tools do not use large sets of complex numerical equations in order to model the atmosphere. Instead, they use cutting edge computational techniques such as optical flow and artificial intelligence algorithms, enabling them to generate useful output at near-instantaneous timescales (Ayzel et al., 2020; Han et al., 2019; Gijben and de Coning, 2017).

Currently, there are a number of state-of-the-art nowcasting systems in operation around the world, typically based in developed countries, which take advantage of large weather radar networks to provide near-instant, wide spread data coverage of precipitation. There are, of course, many operational systems in use globally but Table 1 covers some of the most advanced ones.

| Nowcasting system | Input sources | Region of application | Reference |
|---|---|---|---|
| Short-Term Ensemble Prediction System (STEPS) | Weather radar, NWP | UK | (Bowler et al., 2006) |
| Integrated Nowcasting System through Comprehensive Analysis (INCA) | Weather radar, NWP, satellite, surface station observations | Europe (Alpine regions) | (Haiden et al., 2011) |
| Short-range Warning of Intense Rainstorms in Localised Systems (SWIRLS) | Weather radar, NWP | China | (Srivastava et al., 2021) |
| Auto-Nowcast System (ANC) | Weather radar, NWP, satellite, surface station observations, wind profiler, atmospheric sounding, lightning detector | US | (Mueller et al., 2003) |
| McGill Algorithm for Precipitation nowcasting using Lagrangian Extrapolation (MAPLE) | Weather radar | Canada, US | (Germann and Zawadzki, 2002) |
| Spectral-Prognosis (S-PROG) | Weather radar | Australia | (Seed, 2003) |
| Global Synthetic Weather Radar (GSWR) | Satellite, lightning, NWP | US | (Reen et al., 2020) |

**Table 1.** Information on some of the state-of-the-art nowcasting systems that are currently in operational use around the world.

Weather radar networks are expensive to implement, maintain and often not suitable for regions with mountainous terrain (e.g. the MC), meaning the types of nowcasting systems listed in Table 1 cannot always be implemented. There is, therefore, a widespread interest within nowcasting research in the use of satellite data as the main source of input, especially in the tropics, which can provide constant, widespread coverage of the Earth's atmosphere from space. The advancement of satellite technology in recent years has given us access to data on increasingly

higher spatial and temporal resolutions (e.g. Line et al., (2016) use 1-minute retrievals of 1 km resolution imagery for forecasting), allowing finer detail of cloud structures to be observed and more accurate tracking of weather systems (Sieglaff et al., 2013). This provides the basis for extrapolation nowcasting methods that use the tracked history of weather features (e.g. storms) to calculate motion vectors, which the features are then propagated along to create future predictions (Burton et al., 2022; Vila et al., 2008; Line et al., 2016). The vast volume of satellite

data also makes nowcasting a suitable candidate for the application of machine learning methods to make future predictions of the atmosphere. Most commonly, studies have trained machine learning models to take in consecutive satellite images as input and then output (nowcast) the future consecutive images (Lebedev et al., 2019; Lagerquist et al., 2021).

The MC itself has received little attention in the field of nowcasting, despite the region experiencing regular
intense convective activity affecting the lives of millions. The Indonesian network consists of 42 weather radars



(Permana et al., 2019) but is sparse relative to the size of the country, the country is highly mountainous, and experiences communication issues between sites, meaning real-time full radar coverage of the region is not possible (Permana et al., 2019). One of the radars within the network was used by Ali et al., (2021) to nowcast two rainfall events over southern Borneo. On the other hand, satellite data was used by Harjupa et al., (2022) to apply The Rapidly Developing Cumulus Area algorithm (Sobajima, 2012) to a region of western Java to predict heavy rainfall for 77 events. The limited sample size and domain of these studies makes it difficult to understand how effective the methods are for other regions of the MC. There is, therefore, a need to test nowcasting tools that can be applied and evaluated across the entire MC domain.

pySTEPs (Pulkkinen et al., 2019) is a free, open-source Python library that provides modules for a variety of optical flow-based nowcasting methods (see section 2.1a for optical flow description). The library is designed for use on radar data and has been used to show skilful prediction of stratiform precipitation in the mid-latitudes (Han et al., 2022; Imhoff et al., 2020). To the best of the authors' knowledge, the only study that has applied pySTEPs to satellite data over the tropics is Burton et al., (2022), who produced nowcast skill up to a 4 hour lead time over West Africa. It is this result that motivates the application of pySTEPS to the MC.

This paper presents the evaluation of both deterministic and probabilistic nowcasts produced by applying pySTEPS to satellite data over the MC. The aim is to highlight their strengths and weaknesses and demonstrate their potential use as an operational nowcasting system.

## 2 Data and Methods

### 2.1 Data

This study uses brightness temperature (BT; the temperature a black body would need in order to emit the radiance detected by a satellite) data from the Himawari-8/9 satellites as input to the nowcasting algorithms. Himawari-8 and -9 are passive geostationary satellites with 16 band channels ranging from 0.47 µm–13.3 µm, covering parts of the visible, near-infrared and infrared (IR) spectrum (Bessho et al., 2016). Hourly BT data from channel 13 on-board Himawari-8/9 has been used, which detects IR radiation with a wavelength of 10.4 µm at a spatial resolution of 2 km. Clouds can be clearly identified in BT maps as they have cold tops relative to the surface of the Earth. The data is selected from the December, January and February (DJF) season, which is the peak season for convection over the MC (Birch et al., 2016), for five seasons from 2015/16 – 2019/20 (data retrievals of Himawari-8 began in 2015 (Bessho et al., 2016)).

In this study, nowcasts were produced using three BT maps as input, spaced evenly apart by 1 hour, starting with the latest observation. 3,476 nowcasts were produced for initialisation times every 3 hours from 0000 LT to 2100 LT to incorporate the diurnal variability of weather over the MC, with the number of nowcasts at each initialisation time shown in Table 2. In order to avoid issues of new convection entering the edge of the domain, which cannot be reproduced (optical flow can only propagate convection that exists in the domain at the nowcast initialisation time), the nowcasts were first produced using BT data on a 15°S – 15°N, 90°E – 153°E domain, and then evaluated on a 10°S – 10°N, 94°E – 149°E domain, which still includes the major islands of the MC (Figure 1a).

| Initialisation time (LT) | Number of nowcasts |
|---|---|
| 0000 | 441 |
| 0300 | 422 |
| 0600 | 422 |
| 0900 | 441 |
| 1200 | 440 |
| 1500 | 429 |
| 1800 | 441 |
| 2100 | 440 |

**Table 2.** The number of nowcasts that were produced at each initialisation time throughout the day.



### 2.2 Methods

#### 2.2.1 pySTEPS/optical flow

pySTEPS provides a well-documented framework that allows users to employ optical flow algorithms for both deterministic and probabilistic nowcasting approaches, as well as a range of verification techniques. Optical flow is a computer vision technique that generates velocity fields to describe the apparent motion of objects across consecutive images (Horn and Schunck, 1981). The key assumption of optical flow is that each pixel intensity remains constant across all images as it is advected. Given $I(x, y, t)$ is the intensity of a pixel at time $t = 0$, this results in:

$$I(x, y, t) = I(x, y, t + \Delta t) \, , \tag{1}$$

where $\Delta t$ is the time between image frames. Applying a Taylor series expansion to equation (1) leads to:

$$\frac{\partial I}{\partial x} U + \frac{\partial I}{\partial y} V + \frac{\partial I}{\partial t} = 0 \, , \tag{2}$$

where $U = \frac{dx}{dt}$ and $V = \frac{dy}{dt}$ are the velocity components of the motion field. (2) is known as the optical flow equation. $\frac{\partial I}{\partial x}, \frac{\partial I}{\partial y}$ and $\frac{\partial I}{\partial t}$ can be calculated as they represent the image gradients over space and time, whereas $U$ and $V$ are unknown meaning (2) represents an underdetermined system that cannot be solved directly. Optical flow methods attempt to get round this by applying various spatial constraints to $U$ and $V$. Section b) and c) will describe the two optical flow methods within pySTEPS that were used in this study.

#### 2.2.2 Lucas-Kanade deterministic algorithm

The Lucas-Kanade algorithm (LK; Lucas and Kanade, 1981) is an optical flow method that assumes, for a given pixel, the eight immediately surrounding pixels move along with that given pixel. This assumption results in nine separate versions of (2) (eight from the surrounding pixels and one from the given pixel itself), representing an overdetermined system. A least squares fit method is then applied to the nine equations to obtain the optimum solution for the given pixel. To create a motion field the algorithm first identifies the key features within an image

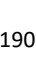
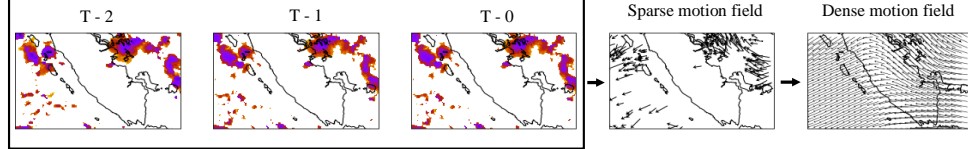

**Figure 2.** An example (using BT over Sumatra) of how a motion field is generated using the LK method. Features are identified and tracked across the three input images (T-0, T-1 and T-2) to generate the sparse motion field. The sparse motion field is then interpolated onto the rest of the domain to produce the dense motion field.

by using the Shi-Tomasi corner detection algorithm (Jianbo Shi and Tomasi, 1994). The velocity field components are then calculated for each feature within the image (using the LK assumption) to create a sparse motion field, which is then interpolated onto the rest of the image (where there are no velocity vectors) to generate a dense motion field. Once the motion field has been determined, the latest observation needs to be advected along the motion field. pySTEPS implements the backward-in-time semi-Lagrangian advection scheme (Germann and Zawadzki, 2002). The nowcast is the resulting advected field. In this study, three consecutive images (the current BT observation and the two images prior) are inputted into the LK algorithm, as shown in Figure 2. Sparse motion



vectors are generated from each successive image pair (T-2/T-1 and T-1/T-0) and then combined together onto one field (sparse motion field). If a pixel in the sparse motion field has two motion vectors associated with it, they are averaged together to produce one motion vector.

The LK algorithm is a deterministic optical flow nowcasting approach that describes the evolution of a field (in this study BT fields) by moving current observations along motion fields. However, this simplistic approach also means it is unable to predict the initiation/growth/decay (IGD) of convection within BT fields.

### 2.2.3 Short Term Ensemble Prediction System algorithm

The pySTEPS library also contains modules for more advanced, probabilistic nowcasting approaches that attempt
to address the IGD problem. The Short Term Ensemble Prediction System (STEPS; Bowler et al., 2006) was jointly developed by the Met Office, UK and the Bureau of Meteorology Research Centre, Australia, and aims to address the issue of unpredictability in the lifetime of convection by injecting fields of varying stochastic noise. It does this by applying a fast Fourier Transform to the current BT field (T-0) to decompose it into cascades of different length scales. Varying intensities of Gaussian noise fields are then injected into each cascade field
depending on the length scale. Cascades containing the small length scale features will receive greater intensity of noise injection, as these features represent the greatest uncertainty in growth and decay of convection. In contrast, the large length scale features receive a lower intensity of noise, as these features represent the least uncertainty in growth and decay. The cascades are then recomposed to produce the new BT field, which is ready for extrapolation. In this work the motion field for extrapolation is generated using the LK algorithm (as in Figure
2, by using T-0, T-1 and T-2 BT fields). Stochastic noise perturbations are also applied to the motion fields to try to capture the uncertainty in the extrapolation of the BT fields. The magnitude of the perturbation increases with respect to lead time as the motion field increases in uncertainty. Finally, the new BT field is extrapolated along the motion field to create one ensemble member of the nowcast. Ensemble members are generated by using new realizations of the noise perturbations to create multiple versions of the nowcast.

### 2.2.4 Verification methods

The stochastic nature of convection in the MC makes it extremely challenging to nowcast the precise location (pixel-to-pixel) of convective activity. When evaluating a nowcasts' ability to predict convection on a pixel-to-pixel basis, the nowcast may be broadly correct but slightly misaligned in location. If we simply take the difference between the nowcast and the verification, this leads to the Double Penalty Problem: firstly, the model is penalised
for a miss and secondly it is penalised for a false alarm in the slightly misaligned location. To overcome this problem Roberts and Lean (2008) developed a method known as the Fractional Skill Score (FSS), which enables a forecast to be verified on a range of spatial scales as opposed to a pixel-by-pixel basis, allowing leeway for minor misalignments. The FSS firstly creates two binary fields from the nowcast field and the observation field by using a threshold value of 235 K (this value was used to try and include the entirety of the convective system
(Machado and Laurent, 2004)) - any pixel with a value below this is set to 1 and any pixel with a value above this is set to 0. A $n \times n$ kernel is then convolved with both binary fields, where $n$ is the desired spatial scale set by the user, and the fraction of pixels within the kernel that have a value of 1 is calculated. The mean squared error (MSE) between the fraction of 1's in the observation kernel, $O_{(n)}$, and the fraction of 1's in the nowcast kernel, $M_{(n)}$, is then calculated:


$$MSE_{(n)} = \frac{1}{N_x N_y} \sum_{i=1}^{N_x} \sum_{j=1}^{N_y} [O_{(n)i,j} - M_{(n)i,j}]^2 \, , \qquad (3)$$

where $N_x$ and $N_y$ are the number of pixels in the longitude and latitude direction. Because $MSE_{(n)}$ is highly dependent upon the frequency of the event it must be compared to the MSE of a relatively low-skill reference
nowcast in order to provide any usefulness, which is defined in Murphy and Epstein, 1989 by:

$$MSE_{(n)ref} = \frac{1}{N_x N_y} \sum_{i=1}^{N_x} \sum_{j=1}^{N_y} [O_{(n)i,j}^2 + M_{(n)i,j}^2] \, . \qquad (4)$$

The final FSS is then calculated as:


$$FSS = 1 - \frac{MSE_{(n)}}{MSE_{(n)ref}} \, . \qquad (5)$$





The nowcast can be evaluated at different spatial scales by changing the value of $n$. In this study 10 km, 20 km, 60 km, 100 km and 200 km were chosen as the spatial scales. This range of scales allows a nowcast to be evaluated in its ability to predict convection on a range of scales. An FSS of 1 can be interpreted as a perfect score whereas an FSS of 0 can be interpreted as a nowcast with no skill. A threshold value for FSS above which a nowcast is useful is given by:

$$FSS_{(useful)} \geq 0.5 + \frac{f}{2} \ , \tag{6}$$

where $f$ is the fractional coverage of pixels with a value of 1 over the entire domain. As $f$ becomes small then $FSS_{(useful)}$ can be approximated by:

$$FSS_{(useful)} \geq 0.5 \ . \tag{7}$$

This is the basic approach to the FSS, which calculates a single score for each nowcast to describe the skill over the whole domain. However, often the skill of a nowcast will vary across the domain due to differences in environments (e.g. land, sea and mountains) and the different interactions that result from these changing environments. Woodhams et al., (2018) developed an adapted version of FSS known as the localised fractional skill score (LFSS), which enables the skill of a nowcast to be evaluated at each pixel across the domain, resulting in a spatial map of FSSs. The LFSS is calculated by adapting (3) and (4) so that instead of dividing the sum of the squared error between $O_{(n)}$ and $M_{(n)}$ over the spatial domain, it is divided over the time domain (i.e. the number of time steps). This results in replacing (3) and (4) with,

$$MSE_{(n)} = \frac{1}{N_t} \sum_{j=1}^{N_t} [O_{(n)t} - M_{(n)t}]^2 \ , \tag{8}$$

and,

$$MSE_{(n)ref} = \frac{1}{N_t} \sum_{j=1}^{N_t} [O_{(n)t}^2 + M_{(n)t}^2] \ , \tag{9}$$

where $N_t$ is the number of time steps.

The skill of a given ensemble nowcast produced by the STEPS algorithm is evaluated by generating the probabilistic nowcast from the ensemble members and then comparing it to the observations at different probability thresholds. At each probability threshold, any pixel in the probabilistic nowcast with a value equal to or greater than the threshold is assigned a value of 1 and any pixel with a value less than the threshold is assigned a value of 0 (creating a binary field). The observation field is then converted into a binary field in the same way except using a threshold value of 235 K. The two binary fields are compared at each corresponding pixel to obtain the number of hits (both pixel values equal 1), misses (nowcast pixel value is equal to 0 but observation pixel value is equal to 1), false alarms (nowcast pixel value is equal to 1 but observation pixel value is equal to 0) and correct negatives (both pixel values equal 0). These metrics are then used to calculate the hit rate [hits/(hits + misses)] and false alarm rate [false alarms/(correct negatives + false alarms)]. This is repeated at multiple probability thresholds and the hit rates are plotted against the false alarm rates to produce a receiver operating characteristic (ROC) curve. The nowcasts with highest skill will minimise the false alarm rate and maximise the hit rate, resulting in a ROC curve in the top left corner of the diagram with a large area under the curve (AUC) score.

Reliability diagrams are used to evaluate how well STEPS's probabilistic predictions compare to the actual observed frequency of events (in this work an event counts as a pixel with a value less than or equal to 235 K). For a given ensemble forecast, the predicted event probabilities are first evenly binned, creating a sub-group of nowcasts for each bin. The frequency for which events are observed is then calculated for each sub-group. The mean event probability within each bin is plotted against the observed event frequency to create a reliability diagram. A perfectly reliable nowcast will predict event likelihoods consistent with the observed frequency i.e. a diagonal X = Y line.



### 3 Results

#### 3.1 Deterministic nowcasting - Lucas-Kanade algorithm

Figure 3e–g shows an example of a nowcast (each lead time is produced using the same T-0, T-1 and T-2 observations) produced by the LK algorithm for a qualitative assessment of the skill against observations (Figure 3a–d), whilst Figure 3h–j provides the LFSS (evaluated on a 20 km scale to show clearly defined differences in skill) at each timestep of the nowcast (where $N_t = 1$) for a quantitative assessment. This particular set of observations contains convection on a range of scales, with regions of propagation and regions of initiation, providing a good example to evaluate the LK algorithm on a range of capabilities (for this reason the same example is used throughout the paper). At T-0, the observed organised, large scale convection e.g. north of Borneo, approximately maintains its shape through T+1 and T+3 and then starts to change in structure at T+6 e.g. east of Sumatra. There is also the development of relatively smaller scale convection observed during the T+3 and T+6 hour lead times e.g. over New Guinea.

Visually, the deterministic LK approach appears to predict the propagation of large scale, organised convection well. The T+1 nowcast best resembles the corresponding observation due to the least amount of new convection developing during this time, as well as little propagation of the organised convection (the nowcast at T+1 closely matches the persistence nowcast). This is seen Figure 3h, which shows high skill in the organised convective regions over the majority of the domain. As the lead time increases, westward propagation of the large scale convection is observed, which appears to be effectively tracked by the LK nowcast at T+3 e.g. east of Sumatra. This is confirmed in Figure 3i, which shows the skill of the nowcast at T+3 remaining high over the regions of organised convection. There is, however, a clear increase in areas of low skill at T+3 and T+6 due to the LK algorithm being unable to reproduce the IGD of convection. At T+6, the change in structure of convection also contributes to the majority of the domain experiencing low skill. The nowcast at this lead time shows the least resemblance to the observations with only the largest regions of predicted convection providing any skill (Figure 3j).

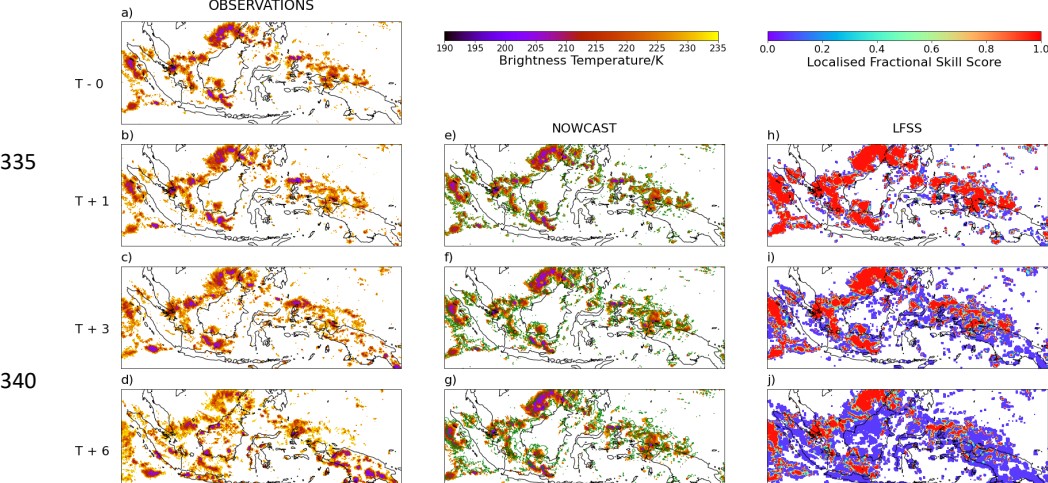

**Figure 3.** a)-d) are maps of BT observations showing convection propagating across the MC on 11 December 2019 starting from the initial observation at 0900 LT (T-0), followed by the observation at 1 hour (T+1), 3 hours (T+3) and 6 hours (T+6) later. e)-g) are the nowcasts produced by the LK algorithm at each of the corresponding observations, and the green lines show the contour of the corresponding persistence nowcast. h)-j) are the LFSS maps produced by evaluating each nowcast against the corresponding observation on a 20 km spatial scale.

Figure 3 highlights some of the key advantages and disadvantages of the LK nowcasting algorithm. Overall, it does well at predicting the propagation of large scale, organised convection. However, because of the principle that underlies optical flow, that each pixel maintains its intensity between timesteps, it is unable to capture the





IGD of convection. Smaller scale convection exhibits higher rates of change in its evolution (Venugopal et al., 1999) and so has the greatest uncertainty associated with it. Initially, this justifies why the majority of low skill is seen at smaller scales (Figure 3h-i). However, at T+6 the difference in small scale features between the observations and the nowcast becomes more widespread and so the low skill spreads further across the domain (Figure 3j).

Figure 4 shows the mean FSSs for all 3,467 LK nowcasts (Table 2) and their corresponding persistence forecasts, evaluated at spatial scales of 10 km, 20 km, 60 km, 100 km and 200 km. The 10 km spatial scale is the smallest scale of evaluation, hence it consistently produces the lowest scores. However, the model still shows good skill on this scale ($FSS \geq 0.5$) at a lead time of 4 hours. Doubling the spatial scale to 20 km increases the skilful lead time by ~1 hour. At the 60 km, 100 km and 200 km spatial scales, the LK algorithm shows skill across all lead 360   times with FSS scores of ~0.54, ~0.61 and ~0.75, respectively at the 6 hour lead time.

Skill reduces with lead time for all spatial scales, and increases with spatial scale at all lead times. On average, the LK nowcasts outperform the persistence forecasts at all lead times and for all spatial scales. Both persistence and the LK nowcasts maintain the same structure and intensity at each lead time (hence relatively smaller skill difference at 1–2 hour lead time), however, the LK algorithm propagates the convection across the domain 365   whereas the persistence remains stationary. This explains the increasing added value of the LK nowcast with lead time, as the observations move further from the persistence nowcast and the skill difference increases. Greater added value of the LK nowcast over persistence is also seen at smaller spatial scales e.g. the skill gap between the persistence nowcasts and the LK nowcasts at the 6 hour lead time is greater for 10 km spatial scale compared to 200 km spatial scale. The trend of FSSs across the shown lead times also varies for different spatial scales. At 370   smaller spatial scales the nowcast is being evaluated on its ability to predict smaller scale convection, which changes most rapidly/unpredictably, resulting in a higher rate of decrease in skill. The FSS evaluation on higher spatial scales smooths out these smaller scale convection changes and so a much lower rate of decrease in skill is seen. The rate of decrease of the FSS evaluated on the 10 km spatial scale appears to show its largest rate of skill decrease at 1 – 2 hours and then decelerates over the following time intervals. On the other hand the rate of skill 375   decrease for the 200 km spatial scale increases with lead time.

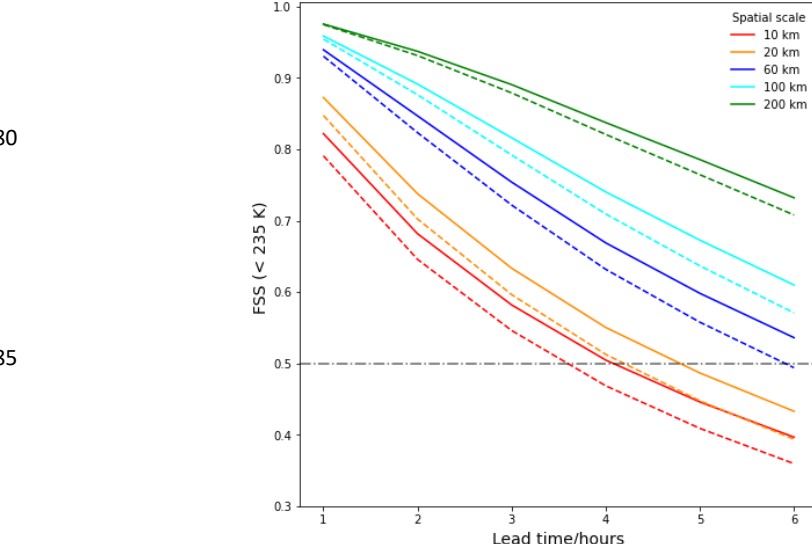

**Figure 4**. Composite FSSs (for 3,457 nowcasts) against lead time for LK nowcasts (solid line) and persistence nowcasts (dashed line), evaluated at a threshold of 235 K for 20 km, 50 km, 60 km, 100 km and 200 km spatial scales. The grey horizontal line marks the 0.5 FSS line, which is considered the cut-off for nowcast skill.



Figure 5 shows the mean LFSS for 3 hour lead time nowcasts over the MC, evaluated at a 100 km spatial scale. Evaluation on this scale has been used as it is able to clearly highlight the variations of skill across the domain.

For all nowcast initialisation times there is consistent noise in the skill over the sea. This may be representative of the stochastic nature of convection initiation over the sea and it would be expected that, if the period of evaluation was extended beyond 2015–2020, the noise would smooth out. During the overnight and morning initialisation times (Figures 5a–d) there is, on average, high skill over the majority of the domain (Figure 5i). This is to be expected as at these times the majority of convection has formed large-scale, organised cloud systems with very

few small-scale convection initiations occurring. These larger-scale systems will most often be propagating offshore of the islands (as part of the diurnal cycle), which the LK algorithm is most effective at predicting.






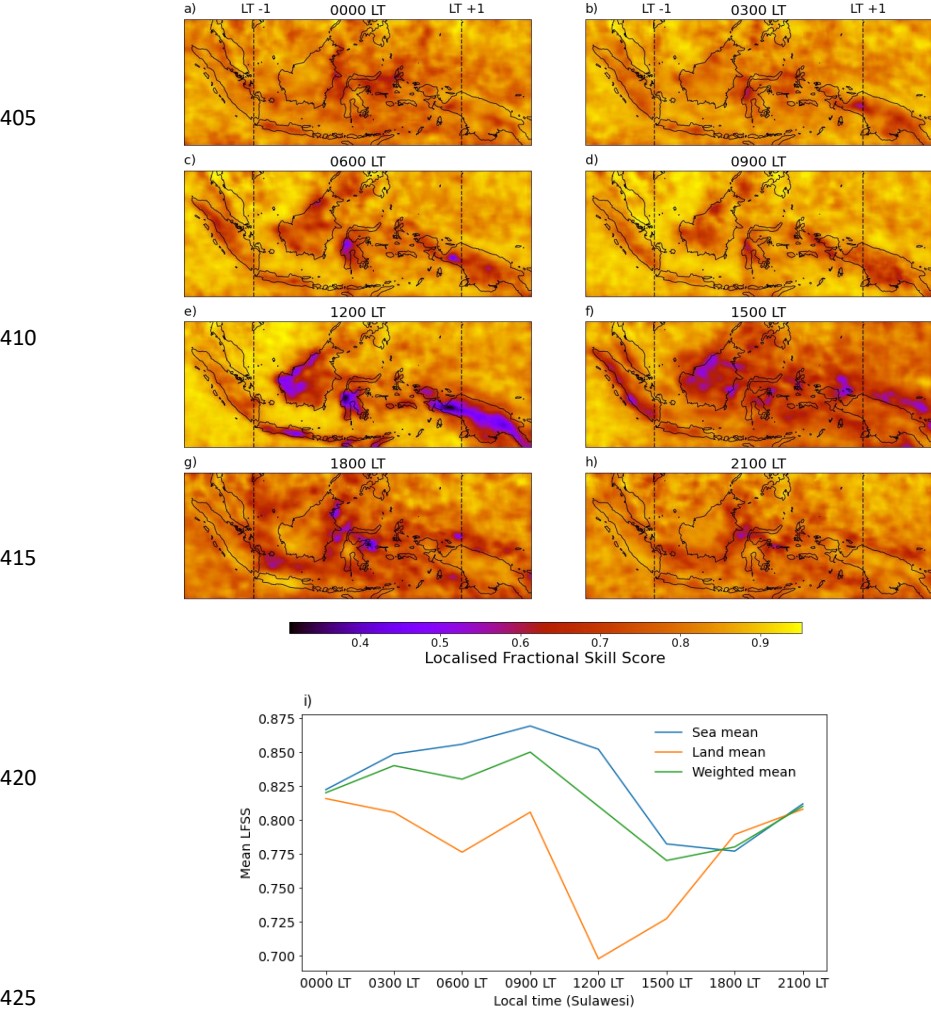

**Figure 5**. a-g) Composite LFSS maps for 3-hour lead time nowcasts that were initialised at 0000 LT, 0300 LT, 0900 LT, 1200 LT, 1500 LT, 1800 LT and 2100 LT, respectively. The LFSS was evaluated on a 100 km scale and the local time is with reference to Sulawesi (the vertical dotted lines show the time difference across the domain). i) The mean LFSS scores at each initialisation time for sea only, land only and the whole domain.

Between 0900 LT and 1200 LT there is a significant drop in the mean skill over land (Figure 5i). Figures 5e–f

show these distinct regions of low skill over land, which are tightly constrained to the coastal and mountainous





regions of the islands and are closely tied to the diurnal cycle of convection over the MC. Convection begins to initiate and develop over the coastal and mountainous regions of the islands in the early afternoon, which is not present at 1200 LT. The LK algorithm is unable to capture this new convection, resulting in low overland skill for the 1200 LT initialisation. Over these locations at this time of day, the LK model would not be a skilful nowcasting

tool. The low skill is still seen at 1500 LT but to a lesser extent. At this initialisation time the T–0 observations that are inputted into the LK algorithm will, on average, contain the majority of the convection that has initiated over the early afternoon. This convection will likely remain stationary over this time but be will be growing in size. Therefore the persisting low skill at 1500 LT is representative of the LK algorithm's inability to predict the growth of convection.

The development of convection starts to slow as storms reach their mature stage in the evening. Less growth results higher skill over the land. At 2100 LT the LFSS map looks similar to the overnight LFSS maps with high skill across the entire domain. Over this 3 hour forecast period the LK algorithm has shown good skill at being able to nowcast the propagation of mesoscale convective systems offshore, which developed overland during the afternoon.

Understanding that the LK algorithm is unable to predict the IGD of convection means that, by identifying anomalous regions of low skill, LFSS maps can be a useful tool for identifying local effects due to land-sea interactions. An example of this is seen in a region of low skill over the Northeast coast of Borneo at 1800 LT. On average, at this time of the day a land breeze begins to develop along the entire concave shaped coastline, potentially causing convergence near the middle. This convergence may then lead to the initiation of convection

that the LK algorithm is unable to predict.

### 3.2 Ensemble nowcasting - STEPS algorithm

Figure 6a provides an example of a 20-member ensemble nowcast (generated using the same T-2, T-1 and T-0 set as for the LK algorithm example in Figure 3), with a lead time of 3 hours, produced by pySTEPS's implementation of STEPS. Over the 3 hour period the main differences between the T-0 and T+3 observations are around New

Guinea where new convection develops over the land and the convection north of the island becomes more scattered (Figure 6b–c). Visually, each ensemble member provides a good prediction of the large scale convection e.g. north of Borneo, with little difference between the members in the predicted shape and structure. The main differences between the ensemble members come from differences in the stochastic fields injected for each member. This is seen in the varying levels of BT intensities in the large scale convection in each member. For

example, the BT intensity over northern Borneo in ensemble member 13 is greater than in ensemble member 2. Differences are also seen between each ensemble member in the distribution of the predicted small scale convection e.g. over the Philippine Sea. Over Borneo, some members have predicted small scale convection which approximately aligns with the new convection observed at T+3 e.g. ensemble member 18, whereas some members have predicted no convection here at all e.g. ensemble member 17.

One of the key aims of STEPS is to address the uncertainty in the evolution of small scale convection. In all members the convection northwest of New Guinea appears much noisier than in the T-0 observation. STEPS has recognized this as a region of uncertainty and addressed it by injecting noise at this scale. When compared to the T+3 observation, it can be seen that the convection does in fact become more scattered and dissipated and so, although STEPS has not been able to precisely predict the new shape of the scattered convection, it has been able

to capture the unpredictable nature of the evolution of this small scale convection.

The 20 member ensemble in Figure 6 has been used to produce the probabilistic nowcast in Figure 7 (extended to 1, 3 and 6 hour lead times). This probabilistic nowcast uses a threshold of 235 K, therefore including all the pixels that were used to produce each ensemble nowcast. At T+1 the probabilistic nowcast shows a high degree of certainty in its prediction of the shape and location of convection, meaning that there is little variance between

ensemble members. The T+1 lead time is the first timestep prediction that the algorithm makes and so it contains the least amount of stochastic noise in the extrapolation motion field, hence the least amount of member variance. As the lead time increases more stochastic noise is injected into the extrapolation motion fields and so the uncertainty of the probabilistic nowcast increases. This can be seen in the reduction in high probabilities over the regions enclosed within the 20% contour. A reduction of high uncertainties within the 20% contour is also matched

with a greater spread of low ensemble probability across the entire domain (outside the 20% contour) at T+3 and T+6.



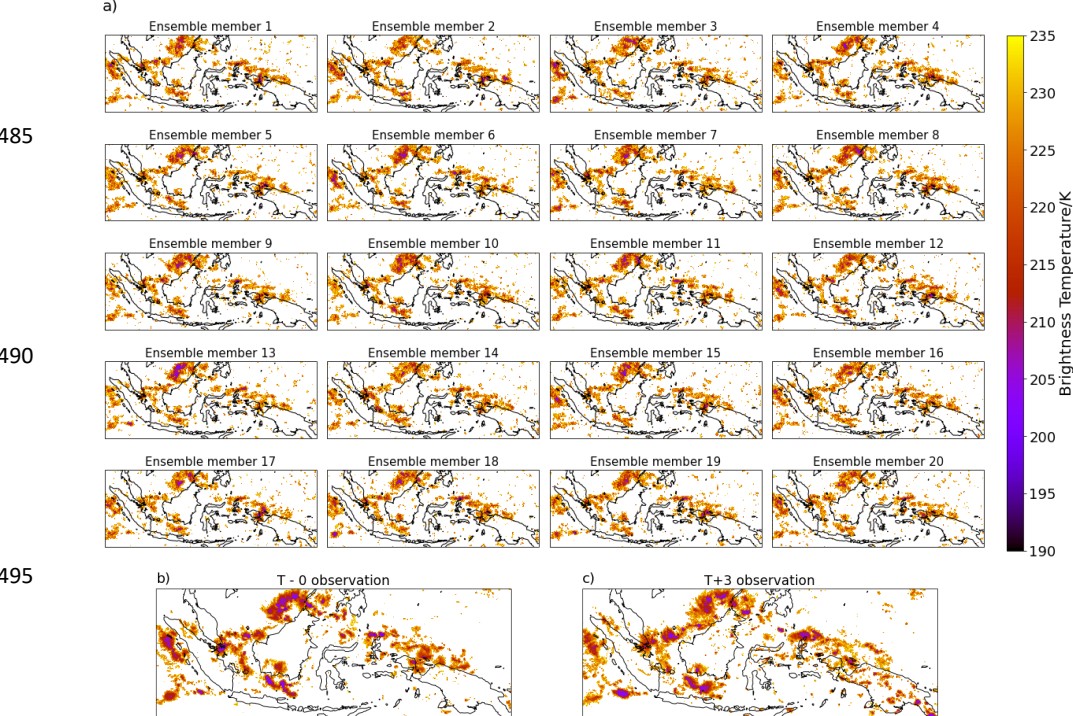




**Figure 6**. a) An example of a 20 member ensemble, 3-hour lead time BT nowcast produced by the STEPS implementation of pySTEPS on 11 December 2019. b) The observation at the nowcast initialisation time and c) the observation 3 hours later.

Figure 7 also shows the number of small scale features in the probabilistic nowcast reducing at longer lead times. For example, At T+1 the convection northwest of New Guinea appears scattered in small blobs, whereas at T+6 STEPS has smoothed out this small scale convection into a larger region. Again, this is evidence of the algorithm's attempt to address the uncertainty in the evolution of small scale convection by replacing it with stochastic noise.

Figure 8a and b show the mean ROC and reliability curves for STEPS nowcasts initialised in the morning (0900 LT) and afternoon (1500 LT), evaluated over the sea and the land. For both surface types and initialisation times, the POD increases at each threshold meaning that, even as more uncertain regions enter the evaluation, the number of hits continues to exceed the number of misses. Furthermore, the POD exceeds the POFD at each threshold indicating that STEPS has skill in predicting regions < 235 K BT over the MC. The greatest POD – POFD difference, which is considered the optimum threshold for a probabilistic nowcast, is shown at the ≥10% likelihood threshold for all of the curves.

For both initialisation times in Figure 8a, STEPS produces higher Area Under the Curve (AUC) scores over the sea (0.80 and 0.78 for 0900 LT and 1500 LT, respectively) than over the land (0.71 and 0.68 for 0900 LT and 1500 LT, respectively), meaning that STEPS has more skill over the sea at these times. This can be explained by lower POD scores over the land, which is due to the STEPS algorithm being unable to capture the new convection that most often develops there (increasing the number of misses). Furthermore, a comparison of each region within initialisation times suggests that STEPS is slightly more skilful in the morning (average AUC of 0.76) than in the afternoon (average AUC of 0.73). The morning – afternoon difference in POFD is also much greater over the land than the sea. This is likely due to a greater decrease in correct negative scores over the land, caused by new convection initiating where there was previously none, which STEPS is unable to predict.









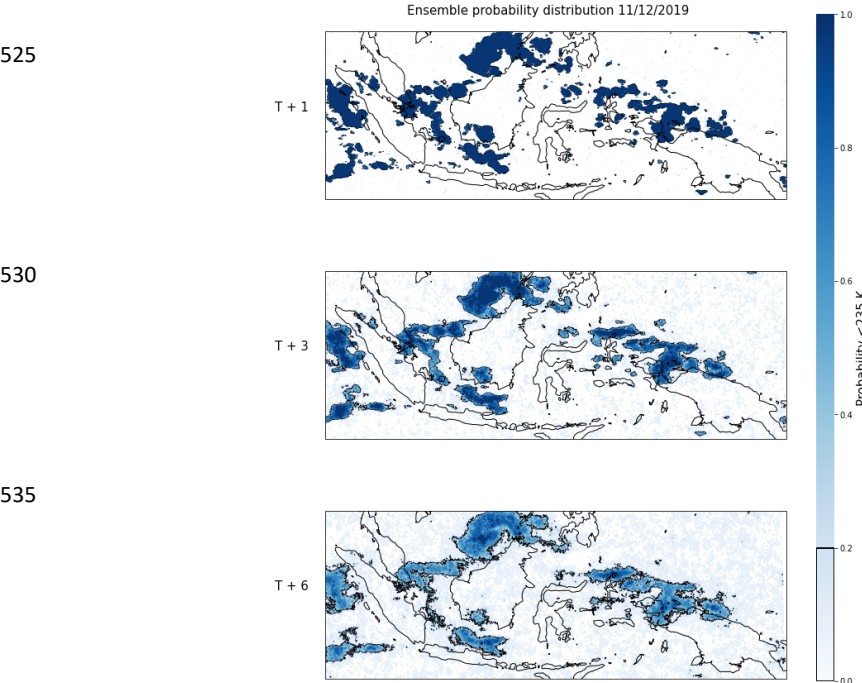

**Figure 7.** An example of a probabilistic BT nowcast for lead times of 1, 3 and 6 hours produced by the STEPS implementation of pySTEPS on 11 December 2019. A BT threshold of < 235 K is used in order to include all pixels in the nowcast.

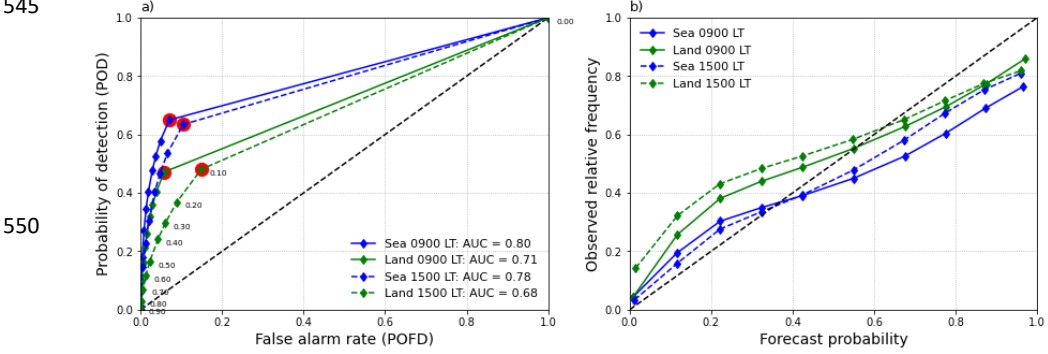

**Figure 8.** Composite a) ROC and b) reliability curves over the sea and land for STEPS 3-hour lead time nowcasts, initialised at 9am (441 nowcasts) and 3pm (405 nowcasts), with a threshold of < 235 K. The numbers next to the green points in a) represent the thresholds used to evaluate the nowcast at different likelihoods.





The positive slope between all the points for each reliability curve in Figure 8b means that, over both surface types, as the observed frequency of events increases, STEPS predicts a higher likelihood of that event occurring. This shows that overall STEPS is able to produce a reliable nowcast for these initialisation times. However, for both regions and times of day, STEPS predicts a lower probability than the observed frequency for probabilities < ~0.35, and a higher probability than the observed frequency for probabilities > ~0.65. The distribution of ensemble predictions is therefore under-dispersive, meaning that the spread of predictions falls within the spread of observations i.e. it does not provide an optimum estimate of uncertainty.

When comparing the two surface types it can be seen that the under-prediction at lower nowcast probabilities is greater over the land, meaning the ensemble distribution has less variance and STEPS is better at capturing uncertainty over the sea for these initialisation times. However, at higher nowcast probabilities, more over-prediction is seen over the sea compared to over the land, meaning that STEPS becomes too confident at predicting higher likelihood events over the sea.

The under-dispersive feature of STEPS over the MC is due to low ensemble member variance, which (as previously mentioned) can be exemplified by visually assessing the lack of diversity between the ensemble members in Figure 6. The main source of ensemble member variance comes from the differences in the stochastic noise fields that STEPS injects into the nowcasts, and so, increasing the range of noise field intensities, or simply adding more members, would likely help to reduce this under-dispersive feature.

### 3.3 Comparison of STEPS, LK and persistence

By applying a threshold to a probabilistic STEPS nowcast, it is possible to produce a deterministic STEPS nowcast (in the form of a binary field) that can be evaluated using the FSS and directly compared to the corresponding LK and persistence nowcast. Figure 9 shows the mean FSSs for 3,467 LK nowcasts (solid), persistence nowcasts (dashed) and STEPS deterministic nowcasts produced using a threshold of ≥10% (dotted; STEPS10). The choice of threshold was based on the results of Figure 8, which show that, for both the morning and evening initialisation times, the optimum likelihood threshold was ≥10%.

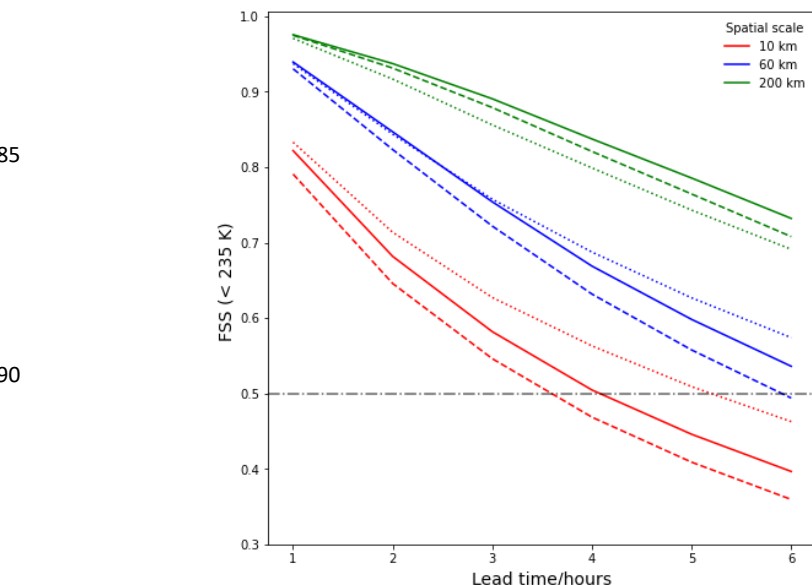

**Figure 9**. Composite FSSs against lead time for 3,457 LK (solid line), STEPS10 (dotted line) and persistence (dashed line) nowcasts, evaluated at a threshold of 235 K for 10 km, 60 km and 200 km spatial scales. The grey horizontal line marks the 0.5 FSS line, which is considered the cut-off for nowcast skill.



At 10 km spatial scale, STEPS10 shows skill up to ~5 hours and has the highest skill across all lead times, outperforming LK and persistence. At 60 km scales STEPS10 still outperforms persistence across all lead times but approximately equals the skill of LK from 1 – 3 hours lead time. Onwards of 3 hours lead time, STEPS10 shows higher skill than LK.

The added value of STEPS10 over both LK and persistence decreases between the 10 km and 60 km spatial scales and, by the 200 km scale, STEPS10 shows the least skill out of all the nowcasts. This is likely due to less of the propagation being detected within the 200 km scale evaluation compared to the 10 km and 60 km scales – hence LK tends towards persistence at greater scales. Unlike LK, STEPS changes the internal structure of the convection through the injection of stochastic noise. This change in the internal structure of convection (as opposed to change due to propagation) will be detected on the 200 km spatial scale and may contribute to a drop in performance relative to LK and persistence.

**4 Conclusion**

A deterministic (LK algorithm) and probabilistic (STEPS algorithm) implementation of the pySTEPS optical flow nowcasting library have been applied to satellite data over the MC to produce nowcasts with lead times of up to 6 hours.

Overall, the LK algorithm predicts the propagation of convection across the domain with skill (FSS≥0.5) up to 4 hours on the 10 km scale (the smallest scale of evaluation) and up to at least 6 hours on the 60 km scale. Similarly, Burton et al., (2022) used the LK algorithm to nowcast convective rain rates over West Africa using retrievals of BTs. Although it is difficult to precisely compare the two sets of results (rainfall retrievals likely have more fine-scale variability and errors in rainfall retrievals will affect nowcast skill (Hill et al., 2020)) they are nevertheless comparable. Burton et al., (2022) show skill up to about 3 hours on the 64 km scale whereas we would expect somewhat higher skill for BT nowcasts over the MC.

Similar to the findings in Burton et al., (2022), the LK algorithm was unable to predict the initiation/growth/decay of convection, which is a manifestation of the optical flow assumption – each pixel maintains its intensity across all timesteps. This inability to predict IGD of convection is clearly seen when analysing maps of LFSS over the MC. Over the sea the LK algorithm shows, on average, good skill at all nowcast initialisation times due to convection being mostly propagating in nature. Over land, however, the model shows high skill in the morning and evening but much lower skill in the afternoon. In the early afternoon this is due to the initiation of convection, which is closely constrained to the mountains. Later in the afternoon the low skill is due to the growth of the convection that initiated earlier on and persists over the mountains. Over the mountainous regions of the MC during the afternoon, the LK algorithm would not be a useful nowcasting tool.

The STEPS algorithm aims to address the issue of unpredictability in the initiation/growth/decay of convection by injecting varying intensities of stochastic noise at different length scales to produce an ensemble nowcast. When analysing a probabilistic nowcast produced by a STEPS ensemble, it can be seen that the injection of noise has a smoothing effect, removing small scale convection and maintaining the shape of the larger, more predictable convective regions.

A composite analysis of STEPS nowcasts using ROC and reliability curves showed that the algorithm is able to produce both skilful and reliable ensemble predictions for 3 hour lead times. When comparing land and sea at different times of day, it was shown that STEPS has highest skill over the sea during the morning (0900 LT initialisation time) with an AUC score of 0.8 (compared to an AUC score of 0.71 over the land). Imhoff et al., (2020) also applied the pySTEPS implementation of STEPS to radar data over the Netherlands to produce nowcasts for a range of lead times. In their work they produced composite ROC curves for nowcasts with a lead time of 95–120 minutes, which had an AUC score of 0.81. The Netherlands experiences far less convective activity than the MC with the majority of its weather coming from propagating frontal clouds. This, therefore, further highlights the effectiveness of STEPS for the MC as it tries to predict convective clouds, which have a more unpredictable nature. However, the disadvantages of STEPS were revealed when analysing the reliability curves. A common feature across both times and regions was the under-dispersive ensemble distributions, which were more extreme over land. This highlights STEPS's inability to predict the low likelihood events e.g. new initiations, and capture the whole uncertainty of the observed system.



To compare STEPS with LK and persistence, a deterministic version of STEPS was produced by thresholding the probabilistic nowcasts at ≥10% (STEPS10). When evaluated, the STEPS10 nowcasts had higher skill at spatial scales of 10 km (across all lead times) and 60 km (from 3 – 6 hours). Therefore, not only does STEPS provide insight into the uncertainty of a convective system, but it can also derive a better single deterministic nowcast than LK and persistence at these scales. However, at a higher spatial scale of 200 km (where relatively less convection propagation is detected), LK nowcasts had the highest overall skill and the injection of stochastic noise produced by STEPS likely caused the STEPS10 nowcasts to have the lowest overall skill.

Continuous nowcasting over the entire MC is a requirement for early-warning systems, which does not currently exist. By providing both nowcast examples and composite nowcast analysis, this paper has shown the effective application of a deterministic (LK) and probabilistic (STEPS) algorithm to satellite data, showing their potential to be used operationally over the whole of the MC. The work highlights the key strengths and weaknesses of both algorithms, providing important information to a potential forecaster using these tools.

*Code availability.* Code is available from the first author upon request.

*Data availability.* Himawari data are available from the ICARE Data and Services Center at the University of Lille, at https://www.icare.univ-lille1.fr

*Author contribution.* JS performed the analysis and wrote the paper. CB, JM and SP participated in technical discussions and provided guidance for the analysis. MB and GP provided guidance for writing the paper.

*Competing interests.* The contact author has declared that none of the authors has any competing interests.

*Acknowledgements.* The author would like to acknowledge the SENSE CDT for providing funding and technical training for this project.

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
