# Peer review of "Applying pySTEPS optical flow algorithms to improve convection nowcasting over the Maritime Continent"

_EGUsphere, 2023_

## Author Response (AR1)

**Paper reviewer comments**

We thank both the reviewers for their time, and for their comments that have improved the clarity of the paper.

**REVEIWER 1**

**Title: to my opinion, the paper does not improve over an existing methodology. It presents the results of the application of the software package on the convection nowcasting over an area. Also it is important to highlight the fact that satellite data are used and not radar data. Therefore the title could be changed to something similar to "Applying pySTEPS optical flow algorithms to the convection nowcasting over the Maritime Continent using satellite data".**

Changed to:
"Evaluating satellite-based convection nowcasting over the Maritime Continent using pySTEPS optical flow algorithms"

**Keywords: The area of application should be added. Also "satellite data" and/or "Himawari satellite"**

Added.

**Table 1. The NWC SAF EUMETSAT Facility could be mentioned.**

Added.

**Line 323: Please define clearly the term "persistence nowcast".**

Added to the end of section 2.2.4:

"In order to gain a useful understanding of the optical flow nowcast skill, a persistence nowcast is used for comparison. A persistence nowcast is considered as having the baseline of minimum skill and is produced by using the latest observation as the next prediction i.e. it assumes that the current weather will persist and be identical at the next nowcast lead time."

**Line 355: Is the term "persistence forecast" interchanged randomly in the text with the term "persistence nowcast"? If no, please define the term "persistence forecast". In general some confusion with the terms "LK nowcast", "persistence nowcast" and "persistence forecast" exists. Please clarify and use with consistency throughout the text.**

Made terms consistent throughout text.

**Lines 396-397: Could you please clarify how the extension of the evaluation period could smooth out the noise over sea?**

Over the sea regions, the stochastic timing and location of convection initiation causes noise in the LFSS field. Extending the evaluation period would increase the number of nowcasted events used in the averaging in equations 8 and 9 (i.e. increase N), and so therefore, it would be expected that the overall field would become smoother. The text has been amended to the following:

"For all nowcast initialisation times there is consistent noise in the skill over the sea. This may be representative of the stochastic nature of convection initiation over the sea. It therefore would be

expected that, if the period of evaluation was extended beyond 2015–2020 (i.e. increasing the number of nowcasted events used in the averaging in equations (8) and (9)), the LFSS noise field would become smoother over the sea."

**Lines 398-400: Could the better scores during overnight and morning hours compared to the rest of the times (especially over land) be attributed to the limited convective activity that minimizes the erroneous propagations too?**

Agreed. The following has been added to the end of the paragraph:

"Furthermore, overnight and during the morning there is, on average, relatively less convective activity than during the day meaning that reduced skill from inaccurate propagation predictions is minimised."

**Lines 509-510: You state at some point that "…the number of hits continues to exceed the number of misses….". However neither POD nor POFD directly suggests that. Overall the ROC curve is a measure the forecast's efficiency to discriminate between the events that actually happened and those that did not. Loosely explained, one could think that it shows how good is a forecast in "finding" the upcoming events without over-forecasting them. Please elaborate on or rephrase your conclusion.**

$Probability\ of\ detection\ (POD) = \frac{hits}{hits+misses}$. In order for the POD to increase, the proportion of hits to misses must increase. The text updated to:

"For both surface types and initialisation times, the POD increases at each threshold meaning that, even as more uncertain regions enter the evaluation, the proportion of hits to misses increases."

The following text added in conclusion to help provide a better understanding of the ROC curve results:

"A composite analysis of STEPS nowcasts was performed using ROC and reliability curves for 3 lead time predictions. The ROC curve is a measure of a nowcast's ability to discriminate between convective events that happened and convective events that did not (the higher the area under a ROC curve the more efficient it is at this), whereas the reliability curve measures how well the probabilistic predictions compare to observations."

**A general comment: Are Himawari BTs offered in a reprojected grid of specific horizontal resolution? If not, it would be beneficial to offer some information on how the satellite data are reprojected, since the skill analysis if provided in specific resolution scales.**

The following has been added to section 2.1:

The full-disc images are transformed to a Cartesian grid with a grid spacing of 2 km, using the gdalwarp command from the Geospatial Data Abstraction Library (Rouault et al., 2023)

Technical comments:

**Line 34: I would prefer "…The "Early Warnings For All" initiative…" instead of "…The Early Warnings For All Initiative…"**

Done.

**Line 24: " …show to…" could be removed.**

Done.

**Line 138:  I would prefer "…ranging from 0.47 μm to 13.3 μm…" instead of "…ranging from 0.47 μm–13.3 μm…".**

Done.

**Line 141: I think that sentence "Convective clouds can be clearly identified…" is more accurate than "Clouds can be clearly identified…".**

Done.

**Line 151-152: Although not necessary, another map presenting the inner and outer domain could be useful.**

Done.

**Figure 4. Could the caption be changed to "Composite FSSs against lead time for 3,457 LK nowcasts (solid line) and…"?**

Done.

**Line 437: I think that "…time be will be…" should change to "time will be..".**

Done.

**Line 440-441: Could it be "…Less growth results to higher skill…" rather than "…Less growth results higher skill…"?**

Done.

**Line 504: "…For example, At T+1…" should read "…For example, at T+1…".**

Done.

**Figure 8: I would suggest to change the horizontal axis legend (and all related text) of the ROC graph to "Probability Of False Detection (POFD)" to avoid confusion with the False alarm Ratio (FAR).**

Done.

**Figure 8: I would strongly suggest to use the same scale for the two axes of the ROC and the reliability graphs (that will result in two "square" graphs).**

Done.

**Line 560: I would use the word "presents" instead of "predicts".**

Done.

**REVEIWER 2**

The manuscript has been well prepared and presented, I will be happy to let it published after a minor revision.

The followings are my comments/suggestions

**The title should be changed to reflect more specifically the content it presented. For example, the manuscript uses satellite data, deterministic and ensemble nowcast, it does not "improve" any current system.**

Agreed. Change to:
"Evaluating satellite-based convection nowcasting over the Maritime Continent using pySTEPS optical flow algorithms"

**This is specifically for MC convective systems using Himawari 8/9, It is better to refer to some previous research with the same topics to decide the BT threshold instead of refer to Machado and Laurent, 2004 for Amazon region. Some more related publications need to be cited to persuade readers about the BT threshold.**

Roca et al., (2017) use a BT threshold of < 235 K to track MCSs over the Tropics. They argue that this threshold value includes both the deep convective and precipitating stratiform regions. A BT threshold of < 233 K has also been used in previous studies for tracking MCSs over the Tropics - Goyens et al., (2012) suggested it as a suitable BT threshold over the Sahel, whereas Fiolleau and Roca, (2013) applied it for tracking over the entire tropics. Feng et al., 2021 used a slightly higher BT threshold value of < 241 K to identify cold cloud systems over Southeast Asia that lasted several hours and reached mesoscale dimensions.

The text was in section 2.2.4 was amended to the following:
"The FSS firstly creates two binary fields from the nowcast field and the observation field by using a threshold value of 235 K - any pixel with a value below this is set to 1 and any pixel with a value above this is set to 0. This threshold was chosen based on previous convection tracking studies in the Tropics (most of which use approximately 233 K – 241 K) and aims to include the entirety of the convective system in the BT image (Goyens et al., 2012; Fiolleau and Roca, 2013; Roca et al., 2017; Feng et al., 2021)."

**Convection in MC in Dec-Jan-Feb is often deep convection. For operational forecast, to have a better forecast a small scale deep convection is more important than large scale organized convection. So that I have some questions and/or suggestions:**
- **Why do the authors use 235K instead of 210K? I would suggest to do an extra calculation for 210K threshold.**
- **What will the FSS, ROC and Reliability curves change if 210K BT used? Will the "core" of convective system be nowcasted better?**

The authors agree with the reviewer that predicting small scale deep convection is an important part of operational forecasting/nowcasting and that these are interesting questions. However, the authors would rather say that the importance of small-scale deep convection versus the "envelope" of large-scale organised convection depends on the application: there are many examples of nowcasting applications that require more than just the "core" of the convective system to be predicted. We use 235 K rather than 210 K to capture the extent of the main precipitating systems rather than just the intense cores (Goyens et al., 2012; Fiolleau and Roca, 2013; Roca et al., 2017;

Feng et al., 2021). The main aim of this paper is to show the advantages and disadvantages of using pySTEPS optical flow algorithms to produce convection nowcasts over the MC, using satellite data, rather than comparing the performance of these algorithms at different satellite data thresholds (representing different scales/intensities of convection). Doing this is not simply an "extra calculation" it requires generating 5 years of thresholded images & nowcasts, and then repeating the evaluation. This would certainly provide interesting future work but is beyond the key aim of this study.

**References**

Feng, Z., Leung, L.R., Liu, N., Wang, J., Houze, R.A., Li, J., Hardin, J.C., Chen, D., Guo, J., 2021. A Global High-Resolution Mesoscale Convective System Database Using Satellite-Derived Cloud Tops, Surface Precipitation, and Tracking. JGR Atmospheres 126, e2020JD034202. https://doi.org/10.1029/2020JD034202

Fiolleau, T., Roca, R., 2013. Composite life cycle of tropical mesoscale convective systems from geostationary and low Earth orbit satellite observations: method and sampling considerations. Quart J Royal Meteoro Soc 139, 941–953. https://doi.org/10.1002/qj.2174

Goyens, C., Lauwaet, D., Schröder, M., Demuzere, M., Van Lipzig, N.P.M., 2012. Tracking mesoscale convective systems in the Sahel: relation between cloud parameters and precipitation. Intl Journal of Climatology 32, 1921–1934. https://doi.org/10.1002/joc.2407

Roca, R., Fiolleau, T., Bouniol, D., 2017. A Simple Model of the Life Cycle of Mesoscale Convective Systems Cloud Shield in the Tropics. Journal of Climate 30, 4283–4298. https://doi.org/10.1175/JCLI-D-16-0556.1

Rouault, E., Warmerdam, F., Schwehr, K., Kiselev, A., Butler, H., Łoskot, M., Szekeres, T., Tourigny, E., Landa, M., Miara, I., Elliston, B., Chaitanya, K., Plesea, L., Morissette, D., Jolma, A., Dawson, N., Baston, D., de Stigter, C., Miura, H., 2023. GDAL. https://doi.org/10.5281/ZENODO.5884351